# Coach Burnout in Relation to Perfectionistic Cognitions and Self-Presentation

**DOI:** 10.3390/ijerph17238812

**Published:** 2020-11-27

**Authors:** Peter Hassmén, Erik Lundkvist, Gordon L. Flett, Paul L. Hewitt, Henrik Gustafsson

**Affiliations:** 1School of Health and Human Sciences, Southern Cross University, Coffs Harbour, NSW 2450, Australia; Peter.Hassmen@scu.edu.au; 2Department of Psychology, Umeå University, SE-901 87 Umeå, Sweden; erik.lundkvist@umu.se; 3Department of Psychology, York University, Toronto, ON M3J 1P3, Canada; gflett@yorku.ca; 4Department of Psychology, University of British Columbia, Vancouver, BC V6T 1Z4, Canada; phewitt@psych.ubc.ca; 5Department of Pedagogical Studies, Karlstad University, SE-651 88 Karlstad, Sweden; 6Department of Sport and Social Sciences, Norwegian School of Sport Science, 0806 Oslo, Norway

**Keywords:** elite sport, coaching, stress, perfectionism, exhaustion, burnout

## Abstract

Coaching athletes is highly rewarding yet stressful, especially at the elite level where media, fans, and sponsors can contribute to an environment that, if not well-managed by the coach, can lead to burnout. Coaches who display perfectionistic tendencies, such as striving for flawlessness, may be particularly vulnerable—even more so if they are overly critical of themselves and have a tendency to ruminate over their performance, or if they are attempting to convey an image of faultlessness, or both. A total of 272 coaches completed a battery of inventories assessing burnout, perfectionistic thoughts, and the tendency for perfectionistic self-presentation. All variables correlated significantly: coaches with higher scores on exhaustion scored higher both on perfectionistic thoughts and self-presentation. However, when three subscales of perfectionistic self-presentation were considered separately, lower and nonsignificant correlations emerged. We believe that this can be explained by the heterogeneous group of coaches participating in this study. Whereas all coaches may at times ruminate privately—self-oriented perfectionism—about their perceived failure to perform to expectations, not all may feel the pressure to present themselves to others as faultless—a more socially prescribed perfectionism. This finding warrants further investigation, preferably comparing coaches at different levels of public scrutiny.

## 1. Introduction

Coaching is associated with many positive facets, but also with the pressure to perform within a stressful environment not known to forgive errors [1]. This may include pressure from media, fans, sponsors, and athletes, which collectively can result in stress for the coach and subsequent burnout [2,3,4,5].

Individuals striving for flawless performance while being overly self-critical of their achievements may display perfectionistic tendencies [6,7,8,9]; coaches are not exempt, and a link between burnout and perfectionism has been reported [10,11]. It is yet to be determined if this wholly or partially explains why coaches withdraw from coaching [12,13]; nevertheless, coaches frequently report suffering from stress, exhaustion, and even burnout [14,15]. The aim of this study is therefore to further explore the relationship between perfectionism and burnout in a group of sports coaches.

Burnout has been argued to be a debilitating stress syndrome, defined by Freudenberger [16] as “a state of fatigue or frustration brought about by devotion to a cause, way of life, or relationship that failed to produce the expected reward” (p. 13). Various instruments have been developed to measure burnout. The Maslach Burnout Inventory (MBI) [17] is frequently used to assess burnout in various occupational settings; it has three subscales measuring emotional exhaustion, depersonalization, and lack of personal accomplishment. Because the factorial validity of the three subscales has repeatedly been questioned [18,19,20,21], we decided to focus on the core element of burnout, namely exhaustion [22]. This also aligns with Freudenberger’s early definition of burnout as a state predominantly defined by fatigue, regardless of whether or not the individual is highly successful [16]. To avoid criticism raised about using instruments such as MBI in sports settings, we decided to use the Coach Burnout Questionnaire (CBQ) as also suggested by Lundkvist and colleagues [20]. This instrument was developed for sports contexts, first in the form of the Athlete Burnout Questionnaire [23] and later modified for coaches [24].

The link between trait perfectionism and burnout has been empirically examined, for example among junior elite soccer players [25,26], intercollegiate varsity student-athletes [27], swimmers [28], and sports coaches [11]. Most studies have been cross-sectional, although efforts to study this relationship longitudinally have been made, for example in a group of junior athletes over three months [29]. The consensus seems to be that perfectionistic striving (e.g., self-oriented striving for perfection, setting high personal standards) is not the main culprit; rather it is perfectionistic concerns (e.g., concern over mistakes, doubts about action, fear of criticism from others) that over time may be detrimental to health and performance [30,31]. The impact of perfectionistic concerns among coaches has also been documented in terms of a link with poor emotion regulation [32]. In their study of sports coaches, Tashman and colleagues [11] distinguished between adaptive and maladaptive forms of perfectionism: conscientious perfectionism was not related to burnout as measured by the MBI, whereas self-evaluative perfectionism was linked with burnout. The authors concluded that this was most likely a result of the appraisal of stressors made by the coaches—the results thus supporting the suggestion that perfectionistic concerns are particularly troublesome [30].

Perfectionistic concerns may lead to rumination—in fact, rumination was the variable most strongly correlated with emotional exhaustion in the study by Tashman and colleagues [11]. Apart from private rumination and concerns about their coaching performance, the public role faced by many coaches may encourage them to manage impressions through self-presentational tendencies. Trait perfectionism reflects the need to be perfect, but some people manage the pressures on them by trying to seem perfect. Perfectionistic self-presentation is a style involving a need to appear perfect while avoiding displays and disclosures of imperfections and mistakes [33,34]. Perfectionistic self-presentation accounts for unique variance in measures of distress beyond the variance attributable to trait perfectionism. Coaches under immense pressure may respond by attempting to consistently convey an image of faultlessness and superior performance. This approach may come at a cost because excessive self-presentational concerns can increase stress, manifesting itself over time in strivings toward bodily perfection, potentially even burnout [7]. To our knowledge, a link between burnout and perfectionistic self-presentation has not been empirically evaluated. The suggestion that chronic stress may cause burnout aligns well with Smith’s [35] burnout model and Lazarus’s [36,37] notion that people failing to cope with stressors over time get more exhausted. The latter may be the result of a discrepancy between perceived demands and available resources—a discrepancy that perfectionists, more than non-perfectionists, may evaluate as being more threatening [11].

Our focus on rumination is in keeping with conceptual advances in the perfectionism field that highlight the need to supplement and extend the current focus on trait perfectionism by incorporating a focus on the cognitive side of perfectionism. Analyses by Flett and associates [38,39,40] suggest that much of the vulnerability of stressed and distressed perfectionists stems from ruminative tendencies. Indeed, the tendency to ruminate about the need to be perfect can be regarded as an internal dialogue that makes the need to be perfect both cognitively salient and a constant source of pressure, especially for those with a negative self-view who see themselves falling short of the ideal of perfection.

We therefore suggest, in line with the above, that coaches who engage in perfectionistic thinking or cognitions are more vulnerable to exhaustion than coaches less prone to this type of rumination. If indeed perfectionistic cognitions result from automatic thought processes that focus on the attainment of perfection or the failure to achieve it, such a finding may explain why perfectionistic cognitions can result in elevated levels of anxiety, depression, general negative affect, and life dissatisfaction [41,42]. Initial evidence has linked perfectionistic rumination with burnout in youth rugby players [43], but the association has not been investigated in coaches.

Our first hypothesis is therefore that coaches actively and frequently engaging in rumination about their own performance, including their attainment or failure to reach perfection, are more likely to score high on exhaustion. Our second hypothesis is that perfectionistic coaches focusing on self-presentational attempts—to either demonstrate their own superiority to others in public situations or to avoid demonstrating imperfection—are particularly vulnerable and therefore will also score higher on exhaustion.

## 2. Materials and Methods

### 2.1. Participants and Procedure

Data were collected with a Web survey where a link was e-mailed to the participants. Two reminders were sent out within two and four weeks. For those working as premier league soccer coaches, the data collection occurred in conjunction with the season ending in November. For high school teachers, the timing was different since they were in different stages in November; however, this was in a period in the middle of the semester in Sweden. Our sample consisted of 272 coaches (231 men and 41 women) with a mean age of 42.9 years (SD 10.2). Of these 272 coaches, 46 were working as soccer coaches in the Swedish Premier League for men and women; 85 were coaching athletics, cross-country skiing, and orienteering at National Sport High Schools; 140 worked as coaches at newly instated National Sporting Programs at the high-school level (athletics, cross-country skiing, orienteering); and one coach failed to provide this information. In total, 36 different sports were represented in the sample: 46% were individual sports (for example athletics, cross-country skiing, alpine skiing, and orienteering) and 54% were team sports (for example football, floorball, ice hockey, and basketball). In the sample, 168 coached full-time (contracted to work 32–40 h per week), 66 worked between 50% and 79% of full-time (20–31 h per week) as coaches, and 40 worked less than 50% of full-time (<20 h per week) as coaches. For coaches working less than full-time in coaching, the vast majority combined their coaching jobs with other positions, predominantly as high school teachers. In total, 254 worked more than 80% (32 h per week) and 18 worked less than 80% (<31 h per week). The mean of actual self-reported working hours in total was 47 h (SD 17.9) per week where 94 persons estimated that they worked more than 50 h per week in total.

### 2.2. Instruments

#### 2.2.1. Perfectionistic Cognition

Perfectionism Cognitions Inventory (PCI) [41] was used to measure thoughts of perfectionism. This instrument comprises 25 items measuring the frequency, using a 5-point Likert scale, of automatic thoughts over the past week indicating the need to achieve perfection (for example, “I can’t stand to make mistakes” and “Why can’t I be perfect?”). Its psychometric properties have been found to be adequate [41,44]. As noted by Flett et al. [39], the PCI has been used in over 40 studies and there is now extensive evidence that the PCI has adequate psychometric properties.

#### 2.2.2. Perfectionistic Self-Presentation

Perfectionistic Self-Presentation Scale (PSPS) [33] with three subscales and a total of 27 items, this instrument aims to measure the individual’s level of Perfectionistic Self-Promotion (PeSP; 10 items, for example: “I strive to look perfect to others”), Nondisplay of Imperfection (NDPI; 10 items, example: “I hate to make errors in public”), and Nondisclosure of Imperfection (NDCI; 7 items, example: “I should always keep my problems to myself”). Participants rated their agreement with the statements using a 7-point Likert scale (1 = Strongly disagree, 7 = Strongly agree); all subscales have shown strong test–retest reliability and validity [33].

#### 2.2.3. Burnout

The Exhaustion subscale from Coach Burnout Questionnaire (CBQ) [23] was used to cover exhaustion, the main symptom of burnout. This instrument with three subscales builds on the original definition of burnout devised by Maslach and coworkers [17,45]. It was later modified to fit athletes (Athlete Burnout Questionnaire) and was subsequently reworded to fit in a coaching context [24,46]. A more recent study shows that CBQ works well in a coaching environment [20]. As argued earlier, the decision to only include the exhaustion factor was made because: (1) exhaustion is seen as the core dimension of burnout [22,23,47], and (2) we predict that striving to create a public image of flawlessness and persistent perfectionistic thoughts will generate distress, which, if prolonged over time, mainly affects exhaustion [7,41]. Exhaustion as measured by the CBQ consists of five items rated on a 5-point Likert-scale (1 = Almost never, 2 = Rarely, 3 = Sometimes, 4 = Frequently, and 5 = Almost always).

### 2.3. Procedure

Ethical approval was first obtained from the Regional Ethical Review Board at Umeå University, Umeå, Sweden. Samples were chosen because of football being the sport with most full-time elite coaches in Sweden and the high school system being the largest employer of coaches in Sweden. A list of coaches was compiled, partly in cooperation with the Swedish Sports Confederation. A mailing list from the Swedish sports federation was used for recruitment of high school coaches and a mailing list from the Swedish football federation was used to reach all coaches in the Swedish Premier League for men and women, as well as the second league for men. Individual e-mails with standardized instructions, including an informed consent form and a link to the Web survey, were sent to all elite soccer coaches active in the Swedish Premier League, and to all coaches employed by the National Sport High Schools or in the National Sporting Programs. After an initial e-mail, three reminders were sent out: the first after one week, the second after three weeks, and the third after another three weeks. A final response rate of 45% was obtained (272 coaches out of 598).

### 2.4. Data-Analysis

A hierarchical multiple regression using SPSS (version 25 IBM Corp, Armonk NY, USA) with maximum likelihood estimation was chosen as the best method to analyze the data, thereby reflecting our aim to determine whether different levels of perfectionism influence the strength of association with exhaustion and at the same time controlling for variables that have previously been shown to impact burnout levels in coaches. The chosen control variables were age, gender, civil status (married or living with partner, or single), coaching level (high school, or Premier League soccer), and total amount of work hours per week.

## 3. Results

### 3.1. Descriptive Statistics

Means, SDs, and Cronbach’s alpha for the five variables are presented in Table 1. Correlations between variables are also shown. The initial analysis revealed statistically significant correlations between all five main variables: higher scores on exhaustion were related to higher scores on perfectionistic thoughts and all dimensions of self-presentation. For the control variables, only civil status had a statistically significant relationship with exhaustion, indicating that single persons score higher on exhaustion.

### 3.2. Multiple Hierarchical Regression Model

The multiple hierarchical regression model is presented in Table 2. The first model containing only control variables explained 6.7% (*p* = 0.005) of the total variance and the second model including all control and predictor variables explained a further 19.6% (*p* F change = 0.000) of the total variance. Of the control variables, three had statistically significant contributions. Coaches who were single scored higher on exhaustion compared with those who were married or living together with a partner; coaches in Premier League football had lower levels of exhaustion than those coaching in high school settings; and coaches who in total worked more hours per week had higher levels of exhaustion. For the predictor variables, two had statistically significant relationships with exhaustion: nondisplay of imperfection from the PSPS (β = 0.21, *p* = 0.007) and perfectionistic cognitions (β = 0.30, *p* = 0.000), which had the largest impact on exhaustion.

## 4. Discussion

The majority of participating coaches displayed low to moderate levels of Exhaustion as measured by the CBQ (see Table 1); this is not surprising given the “healthy worker effect”, suggesting that anyone with high scores has most likely already withdrawn from a situation burning them out [48]. Since the majority were high school coaches, their relatively high job security—compared to elite coaches—could also explain why mean exhaustion levels were low to moderate. The statistically significant correlations presented in Table 1 are therefore particularly noteworthy as restricted variability tends to decrease the magnitude of correlations. The quantile regression analysis also revealed that exhaustion scores were related to perfectionism, or at least to some facets as measured herein and described below.

Firstly, coaches scoring high on perfectionistic cognitions scored higher on exhaustion when compared with coaches scoring lower on the PCI. If perfectionistic thinking is part of an automatic process as has been suggested [41,42], it is likely that these coaches ruminate over their performance, which, in the terminology used by Tashman et al. [11], constitutes self-evaluative perfectionism and is as such considered maladaptive. We expected that there would be a statistically significant association between self-evaluative perfectionism and exhaustion. However, as noted, the analysis showed that the association was statistically significant only for Nondisclosure of Imperfection subscale suggesting that having perfectionistic thoughts, possibly in combination with performance concerns, could be increasingly taxing over time and may result in mounting exhaustion. We then assume that coaches who ruminate do so extensively; it is not something that they can turn on and off at will—it is an automatic process. This is also in line with a form of cognitive bias labelled “all-or-nothing thinking” [49]. Future research should follow up on this suggestion as we do not yet know the extent of variation in rumination or whether there exists a critical level when manageable levels of rumination become exhausting and detectable in inventories such as the CBQ. Because this was a cross-sectional study, we merely acknowledge the existence of a relationship between PCI and Exhaustion, not assuming a causal one. Longitudinal research is therefore warranted to study the role of PCI in exhausted coaches.

Secondly, when age, coaching level, civil status, and gender were controlled, only Nondisplay of Imperfection had a statistically significant association with Exhaustion. The subscales Perfectionistic Self-Promotion and Nondisclosure of Imperfection had no statistically significant relation with Exhaustion. The reason for the (mostly) nonsignificant associations found between Perfectionistic Self-promotion and Exhaustion can, at least partly, be explained by the sample of participating coaches. Because only a minority of the 272 participants were elite coaches (*n* = 45, 17%), it is unlikely that a sufficient number were subjected to scrutiny from media, fans, sponsors, or their own players to an extent that created stress—at least to a level that would result in Exhaustion on a group level. What we do not know, however, is how striving to “create a public image of flawlessness” ([7], p. 16) is affected by “the public”. That is, can club coaches and less publicly scrutinized coaches also become exhausted by trying to uphold images of perfection? Additionally, to what extent do media and fans, in particular, add to the exhaustion created? Future research may want to investigate this. Our results did not suggest that the more public you are as a coach—assuming elite coaches are more public and therefore scrutinized to a greater extent than other coaches—the stronger the relationship between Perfectionistic Self-Presentations and Exhaustion as measured by the CBQ. Rather, it is most likely that the personality trait of being perfectionistic is problematic, regardless of level of coaching. This is also in line with suggestions made by Hewitt and colleagues [33] that striving to create an image of perfection in public situations is the culprit, and this can appear at all levels of competitive sports. However, qualitative interviews have suggested that stepping down from elite to more education-focused coaching can lower perceived exhaustion [14]. Therefore, further studies where coaching levels are more homogenous within the subsamples and more heterogenous between subsamples are suggested to better understand the role of coaching level.

## 5. Conclusions

Conclusively then, we have a somewhat disparate picture emerging between perfectionism and burnout in this study. That is, we detected a significant relationship between Perfectionistic Cognitions and Exhaustion, but largely non-significant relationships between Exhaustion and various measures of perfectionistic self-presentations—although the overall relationship between the PSPS and Exhaustion was significant as expected. The lack of significant relationships between the subscales and Exhaustion can be understood if we consider perfectionistic thinking as a reflection of self-oriented perfectionism—predominantly internally focused—whereas striving to present oneself to others instead reflects socially prescribed perfectionism—something that is more externally focused. Thereby, assuming that self-oriented perfectionism is a constantly present stressor affecting individuals, those affected individuals can hardly escape themselves. Consequently, a coach who ruminates about performance-related issues would do so without the need to interact with anyone, as also discussed by Hill and Curran [30]. Future research is warranted based on suggestions by Flett and colleagues [38,39] that chronic rumination creates a constant source of pressure, as this may explain why exhaustion develops to such a degree that the risk for burnout consequently increases [40]. In contrast, socially prescribed perfectionism would require some form of external interaction, and, if this interaction is intermittent, then the stress level will most likely be lower and its consequences less taxing over time. We believe this to at least partly explain our results, assuming that elite coaches are more concerned about their public appearance than lower-level coaches. Again, this warrants future research comparing coaches at different levels of public scrutiny and their perceptions surrounding this issue, preferably also including qualitative approaches to studying this phenomenon in greater depth. Whether mindfulness-based training could reduce rumination in sports coaches is another avenue of research, as mindfulness has indicated to be effective in elite athletes [50]. Mind–body therapies, such as tai chi, yoga, and qigong, have also shown positive results and can thereby potentially be used for reducing rumination and managing stress levels in sports coaches [51]. In addition, the notion made by Hewitt and colleagues [33] about differences between narcissistic and neurotic perfectionists in relation to self-esteem may also be worth exploring in an elite coaching context in which burnout is prevalent. Whether coach pressure—and a striving to uphold an image of perfection—can contribute to the development of perfectionism in athletes is another line of research worth exploring [52].

## Figures and Tables

**Table 1 ijerph-17-08812-t001:** Means, SDs, and Pearson Correlations (*n* = 272).

Variable	α	M	SD	1	2	3	4	5	6	7	8	9
1. Age	-	42.95	10.16									
2. CS	-	-	-	−0.06								
3. L	-	-	-	0.09	−0.07							
4. WH	-	47.62	17.94	−0.07	−0.10	0.28 ***						
5. Gender	-	-		0.19 **	0.06	0.14 *	0.13 *					
6. EX	0.90	2.10	0.80	−0.10	0.16 **	−0.11	0.12	0.06				
7. PCI	0.93	55.09	16.21	0.27 ***	0.15*	0.08	0.14 *	0.12	0.41 ***			
8. NDPI	0.83	28.61	10.16	−0.14 *	0.05	0.12	0.00	0.05	0.34 ***	0.42 ***		
9. NDCI	0.64	18.31	6.26	−0.12	0.13*	0.07	0.06	0.06	0.34 ***	0.44 ***	0.69 ***	
10. PeSP	0.83	33.72	10.66	−0.12	0.08	0.12	0.09	0.16 *	0.31 ***	0.54 ***	0.67 ***	0.67 ***

* *p* < 0.05, ** *p* < 0.01, *** *p* < 0.001; CS = Civil status (1 = married or living with partner, 2 = single); L = level (1 = high school, 2 = elite football); WH = total working hours; EX = exhaustion; PCI = Perfectionism Cognitions Inventory; NDPI = Nondisplay of Imperfection; NDCI = Nondisclosure of Imperfection; PeSP = Perfectionistic Self-Promotion.

**Table 2 ijerph-17-08812-t002:** Hierarchical Regression with Exhaustion Outcome and Control Variables.

Models		*B*	*SE B*	β
Step 1	(Constant)	1.67	0.38	
	Age	0.00	0.01	−0.07
	Gender	0.15	0.14	0.07
	CS	0.38	0.15	0.16 *
	L	−0.31	0.14	−0.15 *
	WH	0.01	0.00	0.16 *
Step 2	(Constant)	0.41	0.38	
	Age	0.00	0.01	0.05
	Gender	0.03	0.13	0.02
	CS	0.24	0.06	0.10
	L	−0.42	0.13	−0.20 **
	WH	0.01	0.00	0.14 *
	PeSP	−0.05	0.07	−0.06
	NDPI	0.16	0.07	0.21 *
	NDCI	0.10	0.08	0.11
	PCI	0.38	0.09	0.30 **

* *p* < 0.050, ** *p* < 0.001; CS = Civil status (1 = married or living with partner, 2 = single); L= level (1 = high school, 2 = elite football); WH = total working hours; PeSP = Perfectionistic Self-Promotion; NDPI = Nondisplay of Imperfection; NDCI = Nondisclosure of Imperfection; PCI = Perfectionism Cognitions Inventory.

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
