# Peer review of "Coach Burnout in Relation to Perfectionistic Cognitions and Self-Presentation"

_ijerph, 2020, doi:10.3390/ijerph17238812_

Round 1
Reviewer 1 Report
Dear Authors,
Thanks a million for your submission which I found very interesting to read.
Overall, the paper is very well written and presented. However, I would raise the following questions in the hope that it helps you develop the paper further.
L41 – Freudenberger’s definition of burnout refers specifically to not achieving the expected reward, but interestingly, coaches burnout even when they win. Famous examples include Pep Guardiola who took a year out of coaching after his success in Barcelona to recover. This is partially addressed in L46
L108-L121 – Where all the participant soccer coaches? It is not clear, but the Soccer Premier League is mentioned. It would be interesting to understand more about the sports they coached, if not all football. Were there any variations in the results depending on the sport?
L150 – Under procedure, it is described that the list of coaches was compiled with the help of the SSC. Again, it would be interesting to know more about the sample: was it a convenience sample or were there any specific features/criteria sought in the selection process? It is clear that coaches came from 3 distinct groups, but there is no justification as to why these were selected.
L163 – Total amount of work hours. Is this how many hours they were contracted for, or how many actual hours they spent coaching? I would think that the issue with coaches comes when they are working 24/7 or at least thinking about it 24/7. See L179 too. This should be clarified.
L178 – “coaches on in” – delete “on”
L187 – other potential explanations of the low exhaustion levels that may be worth mentioning could be: a) high stability/job security typical of high school-based coaching jobs (did the premier league coaches score differently here?); b) the relative low-level competition these coaches were involved in (by comparison with Olympic coaches or those in top professional leagues). This is explained in L215, but could have been explained earlier. Also, could you have compared the 45 elite coaches to the rest?
L225 As per the above point, when you say that it is most likely the personality trait rather than the level of coaching, could you have compared the 45 elites with the others to see if that held true? (as you suggest in L248-249.
As a general comment, a diagram depicting the negative/positive correlations and their direction would be very helpful as a summary of the findings.
Finally, given that rumination is highlighted as a key factor, could a mention of Mindfulness-Based Training to reduce rumination and research in this area be made in the discusion as a potential recommendation for coaches on the ground?
Again, thanks a million for your excellent paper. I hope these comments are useful.
Reviewer 2 Report
The manuscript is very well written and you have covered an important issue. The introduction provides a clear background about the research and a good rationale of the research is provided. The methods section is well described. Results are systematically presented. However, the discussion should also indicate the strengths of this research. More critical debates should be added.
Reviewer 3 Report
The current study investigated the relationships between coach burnout, perfectionistic cognitions (PCI), and perfectionistic self-presentation (perfectionistic self-promotion/PeSP, nondisplay of imperfection/NDPI; and nondisclosure of imperfection/NDCI) with a sample of professional and high school coaches. The study found that, after controlling for coaches’ individual information, only PCI and NDPI are positively and moderately associated with exhaustion. The current study gave some valuable information about the relationship between coaches’ perfectionism and exhaustion in the Swedish context. However, there are several issues that should be addressed before publication is warranted.
Firstly, it looks like this manuscript is more like a research brief as opposed to a full-length manuscript. Relatedly, the originality of the current study was also somewhat a critical issue because not much new information was introduced in this paper. The author(s) conducted a fair literature review on the given topic, but the reviewer found a hard time to figure out any unique contribution of the current study. For example, in line 55, you criticized that the majority of the related studies have been cross-sectional. However, yours is not different from them. You need to do better work to discuss what the readers should know in terms of the unique contribution of your study. The authors should conduct a more thorough review of the literature. By doing so, a richer and more focused literature review section will allow you to make your research purpose more meaningful to the intended readers as well as to place your findings in a more subtle fashion in and around relevant literature and data.
Secondly, your discussion section was too descriptive – often regurgitating what was reported previously in the result section. Consequently, the authors failed to contribute to the purpose/aim of the current journal. As written, I see almost no contribution to how this study contributes to ‘public health.’ At least, you need to discuss possible intervention strategies, etc.
Thirdly, you need to run a correlation analysis with control variables used in this study, along with the main study variables.
Lastly, the authors need to pay attention to detail. For example, some of the acronyms were never mentioned in the paper, e.g., NDPI and NDCI, as well as how some demographic variables (e.g., coaching level) were coded.
Hope this helps.
Round 2
Reviewer 3 Report
The revised manuscript reads better.
There are a few things to improve before a publication decision.
- briefly report the results of correlations
- The header for "3.1 Multivariate hierarchical models" - you only have one model (so it should be singular) and it is NOT a 'multivariate hierarchical model'. It should be a 'multiple regression model.' You need to have multiple dependent variables to be called 'multivariate'
- Also briefly report regression coefficients of the key predictor variables in writing.
Best of luck.
